# OpenReview forum: "Instructional Segment Embedding: Improving LLM Safety with Instruction Hierarchy"
_NeurIPS.cc/2024/Workshop/SafeGenAi — SafeGenAi Poster_

### Official Review · Reviewer_wiNF · 2024-10-09
**Presents interesting ideas and clear analyses; additional details would be helpful**

**Rating:** 6
**Confidence:** 3

**Review:**

The authors propose a training strategy for LLMs called "Instructional Segment Embeddings" to help models prioritize hierarchical instructions.

Strengths:
- The authors are addressing an important area in LLM safety -- robustness to adversarial attacks -- and show improvement with their method across models and datasets.
- The work is well established in the literature and makes clear connections to various types of adversarial attacks

Weaknesses/Clarifications:
- The authors mention being robust to "harmful requests" which is reasonable.  However, the other side of that coin could be censorship of information.  It would be great to add some nuance to their discussion/presentation of adversarial attacks to include potential mis-use of safety protocols like the one presented.
- Some implementation details are difficult to understand from the paper.  For instance, how is the model told which hierarchical instructions to prioritize?  Is it customizable during training or inference?  It was relatively clear to me how the hierarchy was introduced to the model, but not how the model was told which parts of the hierarchy to prioritize.
- The impact of the training strategy on robustness is shown, but given that the method is designed specifically to impose priority in hierarchical instructions, evaluating changes in how the LLMs prioritized parts of the hierarchy differently could add to the paper.
- Parts of the writing are redundant (e.g., sections 3 and 4) but some of the implementation is still unclear.  A figure showing the training strategy would be very helpful.
- Some things could be better defined to help reader clarity.  For instance, a metric used consistently is "robustness" but how it was calculated s not explicitly defined in the paper.

---

### Official Review · Reviewer_x1fD · 2024-10-10
**This paper proposes a simple yet effective approach to defending against prompt injection attacks. However, additional discussion on potential adaptive attacks would be valuable.**

**Rating:** 6
**Confidence:** 3

**Review:**

This paper introduces Instructional Segment Embedding (ISE), a novel technique to improve the safety of large language models (LLMs) by incorporating instruction hierarchy directly into the model’s architecture. ISE enables the model to prioritize various types of instructions (system, user, data) by embedding this priority information. The empirical results show that this method improves model robustness to adversarial prompts and enhances instruction-following capabilities, with up to 18.68% improvement in robustness and 4.1% increase in instruction-following on evaluated benchmarks.

While the introduction of ISE is a simple yet effective approach, the paper acknowledges its potential limitations against jailbreak attacks. This raises the possibility of other adaptive attacks. During training, the hierarchy is encoded in a specific manner (system, user, data, output). However, if an attacker becomes aware of this structure and introduces attacks that target different hierarchies, the effectiveness of the defense might be diminished. It would be helpful if the paper could expand on this point and discuss potential strategies to address such adaptive attacks.

---

### Official Review · Reviewer_ttdD · 2024-10-12
**Good paper; could improve clarity in writing and include ablation studies**

**Rating:** 6
**Confidence:** 3

**Review:**

The paper introduces Instructional Segment Embedding (ISE), a method to improve the safety and security of LLMs by embedding different types of instructions i.e system instructions, user inputs, and data inputs with distinct segment embeddings. This helps the model differentiate between these input types and ensures that system-level instructions are followed, even if user inputs or data attempt to override them, addressing vulnerabilities like prompt injection, prompt extraction or harmful response. Experiments show improvements in robustness benchmarks as well as on overall instruction following capability.

Pros:
1. The ISE method demonstrates its usefulness in preventing prompt injection and other types of attacks where user prompts or data inputs might be manipulated to override or bypass critical system-level instructions. By embedding segment distinctions into the model, it ensures that system instructions, such as safety guidelines, are not easily overridden by adversarial inputs, making the model significantly more robust in such scenarios.

2. Ease of implementation


Cons

1. Limited Exploration of Instruction Complexity: The paper does not sufficiently explore scenarios involving an increased number of instructions or more complex conflicts between prompts and data inputs. The current experiments focus on relatively simple cases where either the user prompt or external data is manipulated, but not both. A more comprehensive study on how the model performs with multiple conflicting instructions or when presented with weaker, ambiguous system instructions would have provided greater insight into its robustness.

2. No Evaluation with Larger Models: The study is limited to models in the 8B to 13B parameter range, leaving open questions about the scalability of ISE to larger models like GPT-4 or Llama-70B+. Given the complexity of real-world LLM applications, it would have been valuable to see how well this method performs with larger-scale models. Curious to know why the authors excluded larger models when larger versions of llama model are available.

3. Overall question on scalability of approach - how does it perform on increasing levels of system instruction complexity, on larger models with more complex instructions, on multi-turn conversation, on more granular segments (although authors hypothesize it should work) questioning value to real world use cases.

Other feedback
1. Unclear Explanation of the "Hierarchy": While the abstract and various parts of the paper mention hierarchy in terms of prioritization of system instructions, user and data inputs, the implementation seems to only handle segmentation of input types rather than any explicit prioritization or ranking. The mechanism through which these segments are supposed to enforce a hierarchy isn’t clearly explained. The authors do not demonstrate any weighting or formal ranking system that gives system instructions more importance in the model's decision-making process. If it's meant to implicitly prioritize the system prompts, authors should clarify it clearly in the paper.
2. nit: Explain the significance of why the 2 datasets in particular are selected to evaluate ISE